# Improving Code Completion by Solving Data Inconsistencies in the Source Code with a Hierarchical Language Model

**Yixiao Yang**

College of Information Engineering, Capital Normal University, Beijing 100048, China; yangyixiaofirst@outlook.com

**Abstract:** In the field of software engineering, applying language models to the token sequence of source code is the state-of-the-art approach to building a code recommendation system. When applying language models to source code, it is difficult for state-of-the-art language models to deal with the data inconsistency problem, which is caused by the free naming conventions of source code. It is common for user-defined variables or methods with similar semantics in source code, to have different names in different projects. This means that a model trained on one project may encounter many words the model has never seen before during another project. Those freely named variables or functions in the code will bring difficulties to the processes of training and prediction and cause a data inconsistency problem between projects. However, we discover that the syntax tree of source code has hierarchical structures. This code structure has strong regularity in different projects and can be used to combat data inconsistency. In this paper, we propose a novel Hierarchical Language Model (HLM) to improve the robustness of the state-of-the-art recurrent language model, in order to be able to deal with data inconsistency between training and testing. The newly proposed HLM takes the hierarchical structure of the code tree into consideration to predict code. The proposed HLM method generates the embedding for each sub-tree according to hierarchies and collects the embedding of each sub-tree in context, to predict the next piece of code. The experiments on inner-project and cross-project datasets indicate that the newly proposed HLM method performs better than the state-of-the-art recurrent language model in dealing with the data inconsistency between training and testing, and achieves an average improvement in prediction accuracy of 11.2%.

**Keywords:** language model; Hierarchical Language Model; code completion; code recommendation

## 1. Introduction

In recent years, language models have been used in many fields. Many applications, such as machine translation [1], audio recognition [2], and text classification [3] adopt language models to improve model performance. The field of software engineering has recently adopted language models to improve the performance of many tasks, such as code smell detection [4] and software bug detection [5]. To build a code recommendation system, the source code is parsed into a token sequence and the language model is applied to help recommend code snippets for software engineers. Existing works on subjects such as recurrent language models have contributed to the solution of code completion and have helped software engineers to improve the efficiency of developing software.

However, existing state-of-the-art language models are designed specifically for natural languages. A programming language is very different from natural language, and the direct application of language models designed for natural languages to programming languages faces many challenges. One of the biggest differences between programming languages and natural languages is that programming languages have free naming rules. For a user-defined variable or method in a program, different people have different naming habits and rules. This means that it is impossible to use a fixed vocabulary table or dictionary, to include all of the possible variable names or method names in the programming

language. The number of possible words in a programming language is infinite. On the contrary, natural languages often have a fixed vocabulary table, and a dictionary is enough to model all words in such a language. This difference leads to a dilemma when applying language models to code.

Because of the free naming conventions of source code, for different projects, the variable names, method names, or type names in those projects may vary greatly. If the model is trained on one project, it may encounter many unfamiliar words in another project. If the unfamiliar words appear everywhere, the trained language model will be confused and eventually it will not know how to infer the next word. If some words are unfamiliar when making predictions, discrepancies between training and prediction will occur. The discrepancies may gather and explode during a long prediction phase. This phenomenon is also referred to as *exponential bias*, and is the reason for the rapid decline in prediction effect when there exist data inconsistencies between training and prediction. The data inconsistency problem between training and prediction brings about difficulties in applying code completion technology in large-scale industrial scenarios. According to our statistics, this problem is serious when it comes to source code handling. For two different projects with more than 200 files on GitHub, approximately 70% of the tokens in these two projects are different. Thus, the data inconsistency problem needs to be discussed and resolved. By checking a large amount of source code, we discover that the syntax tree of the source code has hierarchical structures, with strong regularity in different projects, and can be used to combat data inconsistency.

As the abstract syntax tree (AST) of source code has a hierarchical structure, in this paper, we propose a novel Hierarchical Language Model (HLM), to improve the robustness of the state-of-the-art recurrent language model, to gain the ability to deal with data inconsistencies between training and testing. For each hierarchy of the code tree, from the bottom of the tree to the root, we use the state-of-the-art language model to accumulate the information of each tree in each hierarchy. Then, the embedding for each tree in each hierarchy is generated. This procedure is named the encoding procedure. In the next step, the code tree is traversed from the root to the bottom, and the existing state-of-the-art language model is used to decode each encountered tree node. This procedure is named the decoding procedure. The decoding procedure takes the embedding for each hierarchy tree generated during the encoding procedure into consideration. It should be noted, that for traditional language models, only the decoding procedure exists. The proposed framework adds an encoding procedure on the basis of the original decoding procedure, making the model more expressive. The encoding procedure actually takes each tree in each hierarchy as a segment. This can help the model to handle long code context, because when a token sequence is long, grouping that long sequence into segments and predicting the next piece of code based on the generated segments, is often helpful.

In summary, to solve the data inconsistency problem between projects, we propose the Hierarchical Language Model (HLM) to generate the embedding for each sub-tree according to hierarchies in the encoding procedure, and collect the embedding of each sub-tree in context, to predict the next piece of code in the decoding procedure. Experiments on inner-project and cross-project datasets indicate that the newly proposed HLM method performs better than the state-of-the-art recurrent language model in dealing with the data inconsistency between training and testing, and achieves an average improvement in prediction accuracy of 11.2%. The main contributions of this paper include the following:

- To the best of our knowledge, this is the first study to specifically discuss the data inconsistency problem of source code and propose a method to specifically solve this problem.
- The proposed method uses the tree hierarchical structure of source code to combat the inconsistency of tokens.
- The new framework divides the single decoding process of the original language model into the encoding process and the decoding process. This proposed framework



can greatly improve the available parameters of the original model and inspire other language models.

- A new tree encoding–decoding mechanism is designed and applied to the hierarchical structures of code.
- Both inner-project and cross-project evaluations are conducted, to compare the performance of models, and an average improvement of 7% is achieved.

## 2. Related Work

### 2.1. Statistical Models for Code Completion

The statistical language model uses the statistical patterns of code to recommend the next piece of code, based on a given code context. The pioneering work in [6], applied the statistical n-gram model to source code to help predict the next piece of code. SLAMC [7] assigned topics to each token to predict the next code token based on tokens in the context and the corresponding topics of these tokens. A large-scale investigation [8] of n-gram models on a large code corpus was conducted and a visualization tool was provided. Cacheca [9] confirmed the localness of the source code and proposed a cache model to improve the code suggestion performance. The pattern of the common application programming interface (API) calls, with the associated parameter objects, were captured by per-object n-grams [10]. The Naive Bayes model was applied to a graph [11] to suggest API usage. A decision tree, together with an n-gram model [12], was applied to solve the problem of code completion. The code was modeled in the form of DSL [13]. Based on DSL, the model was trained in such a way that the model continued sampling and validating until the right code was suggested. For statement-level code completion, the authors in [14] used a self-defined intermediate representation (IR) and a fuzzy search algorithm to search for similar context, to handle the unseen data, in order to improve the n-gram model.

### 2.2. Deep Learning Models for Code Completion

The pioneering work in [15], used a deep language model to solve the problem of code completion, based on the RNN model. A long short-term memory (LSTM) network is a kind of recurrent neural network that introduces the gate mechanism, to capture longer dependencies than the RNN model. The LSTM model was applied to solve the problem of code completion [16,17], in order to achieve higher accuracy. The attention network [18] is applied to the LSTM model to further improve the ability to capture the characteristics of the context. A Pointer network [18], or graph network [19], is adopted to predict the unseen data. The main difference between the two, is that the work in [19] adopts a different switch algorithm and separates the parameters between the language model and repetition learning model. This makes the prediction effect of unknown data more obvious. The work in [20], uses the tree language model with the tree encoding context. To make the token embedding better, the BERT pre-trained method [21] was proposed, to let the BERT model train the token embedding on three general preset tasks and fine-tune the token embedding on specific multi-tasks [22]. Based on the recurrent language model, the graph model [23] was proposed, to capture the long-term dependencies. The already trained GPT2 model [24] was directly used for code completion. The above works do not handle the data inconsistency and the models are not designed to utilize the code structure hierarchies. The proposed method in this paper can be used as a supplement to the above works.

### 2.3. Models for Code Synthesis

Code synthesis involves generating a code snippet based on the hint described in natural languages or other forms. The technology used in code synthesis is similar to the technology used in code completion. Models such as Seq2Seq [25], Seq2Tree [26], and Tree2Tree [27] were proposed for the problem of code synthesis. To synthesize the API sequences based on natural languages, the Seq2Seq model [28,29] was applied. The neural program translation needs to translate one program to another program. The program is in the form of an abstract syntax tree (AST), and the Tree2Tree [30] model translates one

tree to another tree. It should be noted that the proposed model in this paper could also be used as the decoding module in code synthesis tasks. We will further investigate the performance of the Hierarchical Language Model to be used as the decoding module in code synthesis tasks in future work.

### 2.4. Models for Code Classification

The purpose of code classification is to generate an embedding for the whole code, and use the generated embedding to perform classification. The code classification model can be used in the encoding procedure of the proposed framework. The tree-based convolution neural network (TBCNN) [31] applies the general convolution mechanism to the syntax tree of the code, to judge what type of program it is. The convolution-based attention model [32] was applied to help generate the name of a function. TreeNN [33] was adopted for producing representations of homogeneous and polynomial expressions. To categorize expressions according to semantics, based on TreeNN, EQNET [34] additionally adopted an extra abstraction and reconstruction layer, to help capture the semantics of the expressions. Code is organized in statements, and a bi-directional long short-term memory (BiLSTM) model is applied to the embedding of each statement [35] to generate the representation of a code snippet, in order to help classify it.

### 2.5. Hierarchical Language Model for Source Code

To the best of our knowledge, the proposed Hierarchical Language Model (HLM) framework is the first to use the hierarchical structure of code to explicitly handle the data inconsistency between training and testing. The proposed framework is also the first to add an explicit encoding procedure to the original decoding procedure of state-of-the-art language model frameworks. The experiments indicate that the newly proposed model indeed has some special properties.

## 3. Proposed Method

### 3.1. Preliminary

3.1.1. Abstract Syntax Tree (AST)

The code snippet "if $(a + b > 5)$" and its corresponding AST, are illustrated in Figure 1. In this figure, the root is the node with content $if$ and the leaf nodes are node $a$, node $b$, and node 5. Each node in the AST has a content, which is also called a token. In this paper, each node corresponds to a token, and a token must be the content of a node.

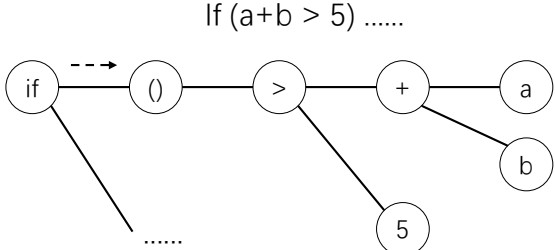

**Figure 1.** An example of an AST.

**Concepts of an AST:** The formal definitions of some concepts of an AST are described here.

- Sibling nodes: In an AST, the sibling nodes have the same parent node. They are also considered to be at the same level or at the same hierarchy. For example, in Figure 1, node 5 and node + are sibling nodes, and node 5 and node + are sibling nodes. Node + is the previous sibling of node 5, and node 5 is the next sibling of node +.
- Descendant nodes: For node $n$, all nodes except node $n$ in the tree rooted at node $n$, are referred to as descendants of node $n$. For example, in Figure 1, the descendant nodes of node > are node +, node 5, node $a$, and node $b$.

- Ancestor nodes: If node $p$ is the descendant of node $q$, then node $q$ is referred to as the ancestor of node $p$.

### 3.1.2. State-of-the-Art Code Completion Method

For an AST, the aim of code completion is to complete each node in that AST one by one. As each node must be predicted one by one, the order in which each node is generated must be decided. We observe that when people write a function, the skeleton of that function is often written first. Then, the details in the skeleton of the function are written, and so on. This order in which the code is written is consistent with the pre-order traversal of AST. Thus, when predicting nodes on an AST, we use pre-order traversal to traverse the tree to predict each encountered node.

**Traditional Language Model Computation Step for Code Completion.** The traditional methods flatten the AST into a token sequence. We assume that the generated token sequence is $t_1$, $t_2$, …, $t_n$. Now, if we want to predict the next token, $t_{n+1}$, based on the existing token sequence, the traditional methods compute the probability $P(t_{n+1}|t_1, t_2, \ldots, t_n)$. Here, we use a simplest recurrent language model to show, in detail, how $P(t_{n+1}|t_1, t_2, \ldots, t_n)$ is computed. The symbol $e_i$ is the embedding for token $t_i$. The embedding $e_i$ for token $t_i$ is just a vector of shape $[1, m]$, where $m$ is the embedding feature size and is set by a human. The shape $[1, m]$, means that the matrix has 1 row and $m$ columns, that is, it is just a vector with $m$ elements. To ease the description, we set $n$ to $i - 1$; now, we want to compute $P(t_i|t_1, t_2, \ldots, t_{i-1})$. With the above definition, the output embedding $h_i$ for predicting $token_i$ is computed as follows.

$$h_i = tanh(W^1 * e_{i-1} + W^2 * h_{i-1}) \tag{1}$$

$h_i$ is generated using $h_{i-1}$ and $e_{i-1}$ with matrix multiplications. In the above equation, $W^1$ and $W^2$ are trainable parameter matrices with shape $[m, m]$. $h_{i-1}$ should be computed recursively until $h_0$, which is a preset trainable parameter vector. Then, $h_i$ is a vector of shape $[1, m]$, which is of the same size as $e_{i-1}$ or $h_{i-1}$. Then, for predicting token $t_i$, we need to compute the probabilities for all possible candidate tokens in the vocabulary. Assuming that there are $v$ total tokens in the vocabulary, given the output embedding $h_i$, the computation step to compute the probabilities for all of these $v$ tokens is as follows:

$$P = softmax(h_i * U) \tag{2}$$

In the above equation, $U$ is a trainable parameter matrix and is of shape $[m, v]$. $P$ is a vector of shape $[1, v]$. Because of the *softmax* operation, each element in $P$ is between $[0, 1]$ and all elements in $P$ sum to 1. When predicting $t_i$, we choose the token with the highest probability in the vocabulary, which means if $P[j]$ is the highest value, then we should choose the $j$th token in the vocabulary as the final recommendation. When training, in the actual existing corpus, if $t_i$ is the the $j$th token in the vocabulary, then $P[j]$ should be the highest value. If $P[j]$ is not the highest value, we should use the random gradient descent algorithm to update all of the trainable parameters, to maximize the $P[j]$.

### 3.2. Differences and Insights between Existing Models and the Hierarchical Language Model

The previous subsection shows that traditional language models handle data points one by one. Even the transformer method still follows the framework described in the previous subsection. The AST shown in Figure 2 can be taken as an example to show the differences between traditional language models and the proposed HLM. If we want to predict the node {}, which is the target of edge $t2$, the traditional language model will handle node $if$, node (), node $>$, node $+$, node $a$, node $b$, and node 5 sequentially, in order to gather the embedding of the encountered nodes and generate the output embedding for predicting the target node. The following equation records the processing path of nodes:

$$Traditional\_Path : if, (), >, +, a, b, 5$$

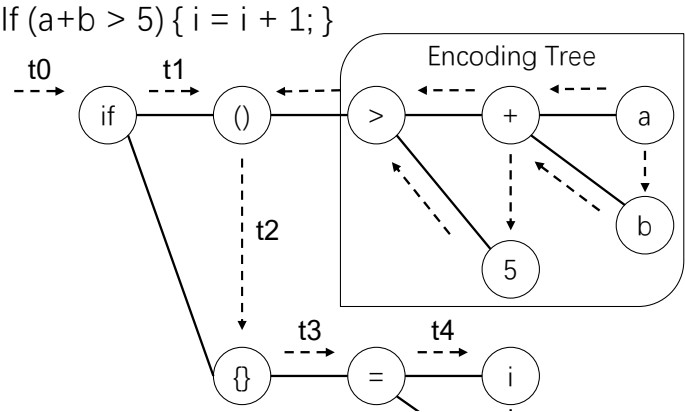

**Figure 2.** Insights of the Hierarchical Language Model.

In the previous subsection, we describe a simple deep language model. However, there are many different language models. Here, we use the abstract symbol $f$, to refer to the procedure of a general language model that takes the information of tokens or the already processed information as the input, and outputs the processed embedding. As can be seen from the following equation, the function f, first handles node $if$ and node $()$. Then, the function f takes the processed embedding generated during the previous step, and the new node $>$, to generate the new embedding. Repeating the handling nodes in the path, the final output embedding $h$ will be generated. With the output embedding $h$, we can use Equation (2) to compute probabilities for all candidate tokens in the vocabulary.

$$Traditional\_Out\_Embedding : h = f(f(f(f(f(f(if,()),>),+),a),b),5)$$

An obvious problem is that precious structural information in the AST will be ignored when the AST is handled in this way. Furthermore, if the nodes are processed in this order, the leaf nodes are processed later than the non-leaf nodes. However, the leaf nodes often contain the names of variables or methods, that vary greatly in different projects. All language models have the following characteristic: the closer the token is to the target, the greater the impact on the prediction result of the language model. Thus, we cannot let a token that varies greatly between different projects be processed. The insights of the design of the proposed Hierarchical Language Model, are that the AST should be processed in hierarchies and that it should not be processed last.

Thus, when predicting the node $\{\}$, which is the target of the edge $t2$, the Hierarchical Language Model (HLM) will traverse the AST and collect the information of complete trees in different hierarchies. For example, node $>$, node $+$, node $a$, node $b$, and node 5 in Figure 2, make up a complete sub-tree. The models designed especially for trees can be used to generate embedding for that tree. The tree embedding is generated in post-order traversal, meaning that node $a$ and node $b$ in the bottom hierarchy are handled first; then, node $+$ and node 5 in the second bottom hierarchy are handled later. It should be noted that if we want to predict the target of edge $t2$ in Figure 2, the target node being predicted is the descendant of node $if$, so we assume the tree rooted at node $if$ is not complete. Similarly, the target node being predicted is the next sibling of node $()$, so we also assume the tree rooted at node $()$ is not complete. For nodes which are not in a complete tree, we use the same method as the traditional language model to handle these nodes. So, node $if$ and node $()$ are handled first, in the pre-order traversal of the AST, to generate an embedding, named $embed_{pre\_order}$. Then, node $>$, node $+$, node $a$, node $b$, and node 5 are handled in post-order traversal of the AST, to generate another embedding, named $embed_{post\_order}$. To generate $h$, to be used for computing the probabilities of all candidate tokens in the vocabulary in Equation (2), we can design a function f, such that $h = f(embed_{pre\_order}, embed_{post\_order})$. The details will be described in the next subsection. The procedure to generate $embed_{pre\_order}$ is named the decoding procedure. The procedure to generate $embed_{post\_order}$ is named the

encoding procedure. By doing so, the original single decoding procedure is divided into two procedures, and in the encoding procedure, we can use models specifically designed for trees, to generate embedding. The following equation records the processing path of HLM.

$$Decoding\_Path : embed_{pre\_order} = f^1(if,())$$

$$Encoding\_Path : hierarchy_1 = f^2(a,b)$$
$$hierarchy_2 = f^2(+,5,hierarchy_1)$$
$$embed_{post\_order} = f^2(>,hierarchy_2)$$

$$Final\_Out\_Embeding : h = f^3(embed_{pre\_order}, embed_{post\_order})$$

### 3.3. Hierarchical Language Model

The encoding procedure of HLM, is to generate the embedding for each tree (sub-tree) in the AST. The decoding procedure of HLM, is to accumulate the information of context (consists of sub-trees) to predict the next node. For the proposed Hierarchical Language Model (HLM), to predict node n at the specified position, HLM accumulates information based on the *decoding path* of node n.

### 3.3.1. Encoding Procedure of HLM

We use post-order traversal to traverse the AST, from the leaves to the root, in order to encode a tree. For a leaf node $n$, we assume its token embedding is $\tau_n$, and the encoding $(cell_n^{enc}, h_n^{enc})$ of the tree rooted at node $n$ is computed by Equation (3). In Equation (3), $cell_0^{enc}$ and $h_0^{enc}$ are preset trainable parameter vectors.

$$cell_n^{enc}, h_n^{enc} = LSTM(\tau_n, cell_0^{enc}, h_0^{enc}) \tag{3}$$

For a non-leaf node $m$, we assume its token embedding is $\tau_m$, and its children are put in a list named $m\_children$. The encoding $(cell_m^{enc}, h_m^{enc})$ of the tree rooted at node $m$ is computed with Equation (4). In Equation (4), $cell_0^1$, $h_0^1$, $cell_0^2$, and $h_0^2$ are preset trainable parameter vectors. $h_{mc}^{enc}$ is the encoding for the tree rooted at the child node $mc$ of node $m$. It should be noted that $h_{mc}^{enc}$ should be generated in the same way as $h_m^{enc}$.

$$
\begin{aligned}
&cell_{cf}, h_{cf} = cell_0^1, h_0^1 \\
&cell_{cb}, h_{cb} = cell_0^2, h_0^2 \\
&for\ mc\ in\ m\_children : \\
&\quad cell_{cf}, h_{cf} = LSTM(h_{mc}^{enc}, cell_{cf}, h_{cf}) \\
&for\ mc\ in\ reverse(m\_children) : \\
&\quad cell_{cb}, h_{cb} = LSTM(h_{mc}^{enc}, cell_{cb}, h_{cb}) \\
&cell_m^{enc}, h_m^{enc} = 2DimensionalLSTM(\tau_m, cell_{cf}, h_{cf}, cell_{cb}, h_{cb})
\end{aligned}
\tag{4}
$$

We use the AST in Figure 3 to show the computation steps for the non-leaf node P, of which the number of children is three. $cell_P^{enc}$ and $h_P^{enc}$ are computed in Equation (5).

$$
\begin{aligned}
&cell_{cf}, h_{cf} = LSTM(h_{S3}^{enc}, LSTM(h_{S2}^{enc}, LSTM(h_{S1}^{enc}, cell_0^1, h_0^1))) \\
&cell_{cb}, h_{cb} = LSTM(h_{S1}^{enc}, LSTM(h_{S2}^{enc}, LSTM(h_{S3}^{enc}, cell_0^2, h_0^2))) \\
&cell_P^{enc}, h_P^{enc} = 2DimensionalLSTM(\tau_P, cell_{cf}, h_{cf}, cell_{cb}, h_{cb})
\end{aligned}
\tag{5}
$$

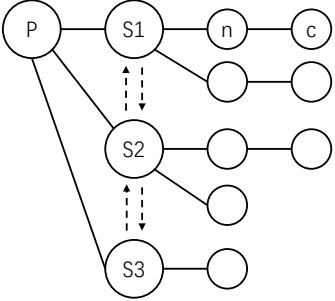

**Figure 3.** Encoding of the AST.

3.3.2. Decoding Procedure of HLM

**Decoding Path of HLM:** The *decoding path* of HLM, for node n, is the transfer path from the root to node n. From a node, only the first child of that node or the next sibling of that node can be transferred to. Thus, the candidate transfer paths of the AST in Figure 1 are shown in Figure 4. In Figure 4, the solid arrow represents the transition to the first child, and the dotted arrow represents the transition to the next sibling node. Thus, under this definition, a directed acyclic graph (DAG) was generated, and the transfer path from the root to each node was uniquely determined. In detail, from the root node of the tree, if node n is the descendant of the root node, to reach node n, we must transfer from the root node to the first child of the root node. After reaching a new node, then, if node n is the descendant of the newly reached node, we must transfer to the first child of the newly reached node. Otherwise, we must transfer to the next sibling of the newly reached node. If we continue transferring in this way, we finally reach node n. There are two kinds of transitions: transfer to first child and transfer to next sibling.

If (a+b > 5) { i = i + 1; }

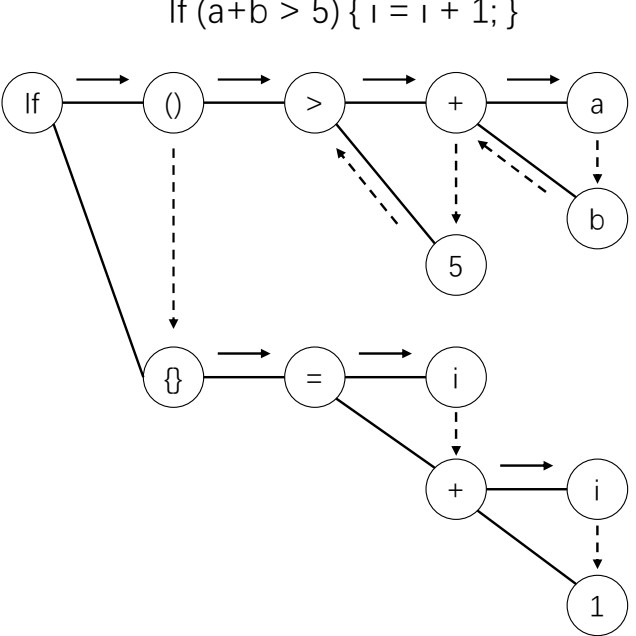

**Figure 4.** Path from root to node.

**Transition on Decoding Path of HLM:** As described above, the *decoding path* consists of a sequence of transitions. In Figure 5, the dotted arrows give an illustration of the path and the transition from the root to node + (the second child of node =). Each transition between nodes on the path is marked as $t_0, t_1, t_2, \ldots t_5$. The information flow of a transition represents the accumulated information of previous transitions before that transition. The information that flows for each transition has a fixed data format: (*cell, h*); *cell* and *h* are

two feature vectors of fixed length. The symbols $cell_i$ and $h_i$, represent the information on transition $t_i$. Note that for each transition $t_i$, the source node of that transition is named $src_{t_i}$, and the target node of that transition is named $tgt_{t_i}$. For node $src_{t_i}$, all descendant nodes are named $src_{t_i}^{descendants}$.

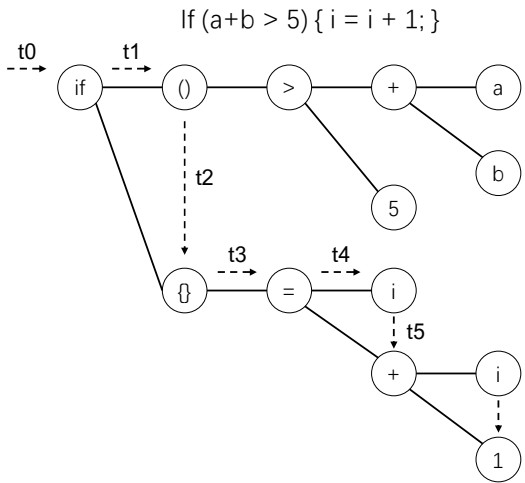

**Figure 5.** Candidate paths of the AST.

**Detailed Decoding Step of HLM:** Then, we iterate the transitions one by one, to compute the accumulated information for predicting node n. At first, the information of transition $t_0$: $cell_0$, $h_0$ is set to a fixed default value. Then, for each transition $t_i$, if $t_i$ is of the "transfer to the first child" type, we use Equation (6) to compute the information for transition $t_i$. We assume that the token embedding of the source node of the transition $t_i$ is referred to as $\tau_i$. The information of transition $t_i$ is computed by

$$cell_i, h_i = LSTM(\tau_i, cell_{i-1}, h_{i-1}) \tag{6}$$

The encoding of the tree rooted at the node $src_{t_i}$, in the encoding procedure described in the previous subsection, is referred to as $cell_{src_{t_i}}^{enc}, h_{src_{t_i}}^{enc}$. For the encountered transition $t_i$, if $t_i$ is of the "transfer to the next sibling" type, the information of transition $t_i$ is computed by

$$cell_i, h_i = 2DirectionalLSTM(cell_{src_{t_i}}^{enc}, h_{src_{t_i}}^{enc}, cell_{i-1}, h_{i-1}) \tag{7}$$

The computed $cell_i$, $h_i$ can be used to predict the target node $tgt_{t_i}$ of transition $t_i$. With the generated $h_i$, Equation (2) can be used to compute the probabilities of all candidate tokens, and the top-k tokens with the highest probabilities are taken as the final results.

## 4. Experiment

The Hierarchical Language Model can be applied to any programming language that can be parsed into an abstract syntax tree. As the number of projects written in the Java programming language is the largest in Github, in this experiment, famous Java projects are downloaded from Github to enter into the code corpus (dataset) for experiments. The source code of each downloaded project is pre-processed to ensure its quality. For each code corpus, the training set accounted for 60%, the validation set accounted for 15%, and the test set accounted for 25%. In experiments, the validation set is used to prevent over-fitting. Every function is parsed into AST and every AST is regarded as a training example or as a test example. Each node in an AST is predicted. The models in the experiments are trained to predict each node in the AST correctly. The accuracy is the summation of the prediction accuracy of each node in each AST. Some sequential models, such as RNN or LSTM, cannot be directly applied to data in tree structures. IThe tree will be flattened into a sequence, making sequential models applicable.

*4.1. UNK Setting*

In natural language processing, the least frequently occurring words are marked as unknown word ($UNK$). In order to avoid $UNK$ being the most frequent words, we set the least frequently appearing words in the training set as $UNK$. Thus, $UNK$ can still be rare words, but not the most frequent words. In test data, the embedding of non-vocabulary words is replaced with the embedding of $UNK$, but we do not think $UNK$ is the right word when computing prediction accuracy.

*4.2. Datasets*

In this experiment, three datasets are generated. Datasets 1 and 2 are the inner-project code corpus. Dataset 3 is the cross-project code corpus. Table 1 shows the composition of each dataset. Dataset 1 consists of Java files in the main module of project *apache commons lang*. The size of Dataset 1 is 2.8 MB. The *apache maven* is a famous project; we downloaded the source code from its official website (not on GitHub). The size of the project is 4.4 MB. As observed from open-source projects, many files contain a large amount of Java documents, comments, or small functions, with only one or two statements. Those noisy data should be removed. For generating cross-project datasets, we used the following three steps, to generate high-quality datasets containing long and non-noisy code. The first step was to choose two to four projects from Github at random. The second step was to compute a score for each Java file in each project: the total number of nodes in functions divided by the total number of functions, resulted in the score for a Java file. Given the threshold for the size of the dataset (for example 8 MB), the third step is to select the top Java files with the highest scores in each project, to mix into a dataset until the threshold for the size of the dataset is reached. Dataset 3 contains the top-scored Java files from projects *Gocd* (5023 stars), *apache-incubator-joshua* (73 stars), *vlc-android-sdk* (723 stars), and *locationtech-geowave* (344 stars). In all datasets, functions with less than 100 AST nodes or more than 10,000 AST nodes were removed. The evaluation results on Dataset 3 are more convincing, because such results reflect the performance of the model on different projects. The last column in Table 1, shows the vocabulary size of each dataset.

**Table 1.** Datasets.

|  | **From Projects** | **Size** | **Vocabulary** |
|---|---|---|---|
| Dataset 1 | apache commons lang main | 2.8 MB | 1273 |
| Dataset 2 | apache maven | 4.4 MB | 5283 |
| Dataset 3 | gocd, apache-incubator-joshua and locationtech-geowave | 7.53 MB | 8030 |

*4.3. Baselines*

To evaluate the performance of our model, some baselines needed to be trained. RNN and LSTM were taught to predict the next token based on already predicted tokens in the sequence generated by flattening a tree. RNN and LSTM are classical models for capturing patterns in sequential data. These two models are included in the baseline. Compared with RNN, LSTM has a more powerful ability to capture the long-term dependency in sequential data. Every model in the baseline needs to predict every token, for every function in the dataset.

*4.4. Hyperparameters*

We used the Adam algorithm to compute the gradients. We trained examples one by one, instead of grouping examples into batches, because different ASTs may have different numbers of nodes. The vector size for the feature vector of one token was 128. All other parameters were decided successively.

*4.5. Termination Condition*

We used the strategy of early stopping for the termination of model training. The model stopped instantly if the the prediction accuracy on the validation set started to decrease.

*4.6. Platform*

The experiments were conducted on a desktop computer. The CPU of the computer was i5-8400, the GPU was a Geforce RTX 2070, and the memory size was 32 GB. The implementation of the model was based on TensorFlow.

*4.7. Evaluation*

The metrics to evaluate the performance of different models in this experiment include top-k (top-1, top-3, top-6, and top-10) accuracy and mrr (mean reciprocal rank). The top-k accuracy is computed by judging whether the right token appears in the first k recommendations of the code completion model. When predicting the token of the next node, if the oracle token appears in the $r$th recommendation, then the rank of this recommendation (completion) is r and the reciprocal rank of this recommendation is $\frac{1}{r}$. The mrr is computed by averaging the reciprocal rank of the oracle token for each code recommendation (completion).

Table 2 shows the top-k accuracy and the mrr evaluated for four datasets. HLM refers to the Hierarchical Language Model. On all three datasets, RNN performs the worst. On small datasets, i.e., Dataset 1 and Dataset 2, HLM achieved, on average, an 8.5% higher top-1 accuracy than LSTM. On large datasets, i.e., Dataset 3, HLM achieved, on average, a 16.9% higher top-1 accuracy than LSTM. After carefully investigating the experimental results, we found that the proposed HLM is good at predicting tokens related to the syntax tree structure. For top-1 accuracy, the different models used in the experiments had a large degree of discrimination. For top-3, top-6, and top-10 accuracy, the distinction between different models became smaller and smaller. This illustrates that the top-1 prediction accuracy is the most convincing. From the perspective of top-1 prediction accuracy, HLM performs better than other models. Figure 6 shows the top-1 accuracy for different models on different datasets. As can be seen, the proposed HLM achieves the best results for all datasets. This figure intuitively shows the effect of the proposed model.

**Table 2.** Evaluation results on test sets.

| DS | MD | Top-1 | Top-3 | Top-6 | Top-10 | mrr |
|----|------|--------|--------|--------|---------|------|
|   | RNN | 32.8 | 45.1 | 53.6 | 59.8 | 0.41 |
| 1 | LSTM | 46.8 | 60.7 | 67.2 | 71.3 | 0.55 |
|   | HLM | **50.3** | 64.7 | 70.7 | 73.8 | 0.58 |
|   | RNN | 44.0 | 56.9 | 63.8 | 69.3 | 0.52 |
| 2 | LSTM | 50.9 | 64.1 | 69.6 | 72.5 | 0.58 |
|   | HLM | **55.7** | 69.5 | 73.4 | 75.2 | 0.63 |
|   | RNN | 34.8 | 52.2 | 56.4 | 59.3 | 0.43 |
| 3 | LSTM | 48.6 | 61.3 | 68.6 | 70.4 | 0.56 |
|   | HLM | **56.8** | 64.9 | 71.3 | 72.5 | 0.62 |

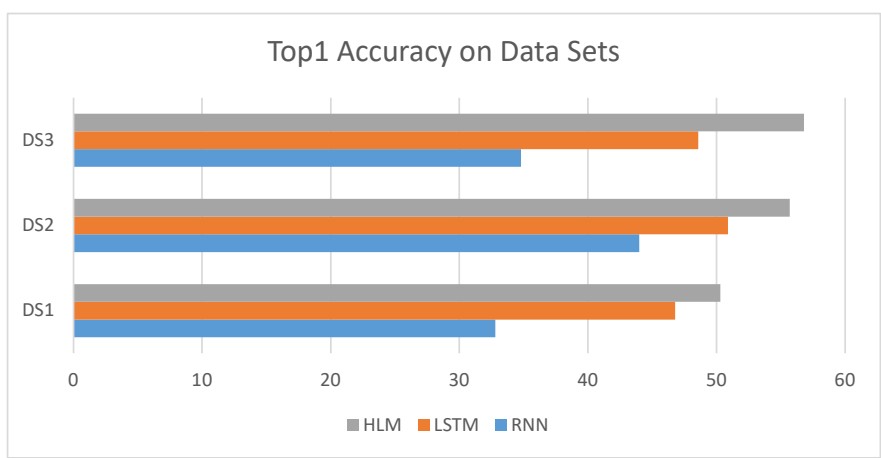

**Figure 6.** Top-1 accuracy for different models.

Table 3 shows the training time for the different models on different datasets. As the proposed model splits the original decoding procedure into encoding procedure and decoding procedure, the training time for one round is twice as long as the baseline model. However, as the proposed model captures the characteristics of code structures, the proposed model is much easier to converge than the baseline model, which does not capture the code hierarchy structures. The number of rounds required for the convergence of the new model is generally one half to one third of that of the baseline model. In total, the proposed framework still uses less time to complete the training procedure than the baseline model.

**Table 3.** Training time for different models.

|       |      | Time for One Round | Number of Rounds to Converge | Total Time |
|-------|------|--------------------|------------------------------|------------|
| DS1   | LSTM | 1 min              | 18                           | 18 min     |
|       | HLM  | 2 min              | 7                            | 14 min     |
| DS2   | LSTM | 6 min              | 29                           | 174 min    |
|       | HLM  | 15 min             | 9                            | 135 min    |
| DS3   | LSTM | 22 min             | 35                           | 770 min    |
|       | HLM  | 40 min             | 11                           | 440 min    |

As can be seen from the experimental results, the proposed model achieves the best results. By introducing more parameters in the proposed framework, the model can fit the data better. Furthermore, in the encoding procedure, the tree model is adopted, to capture the characteristics of the abstract syntax tree (AST) of the code, which also contributes to the improvement. The HLM uses post-order traversal to traverse the AST, to encode all subtrees. This kind of encoding is good at handling unseen data. As the unseen token is often on the leaf node of the AST, if we continue to abstract important information from the leaves to the root, the impact of unseen data on leaves is often reduced. Meanwhile, the standard LSTM model treats all tokens equally. When encountering unseen tokens, the standard LSTM model handles the unseen token, and information bias appears. The information bias can be accumulated if there are many unseen data in a long sequence. This problem is called *exponential bias*. The ability to reduce the impact of the unseen tokens is the key to performance improvement. In summary, the top-1 prediction accuracy on the test set is representative, with HLM performing better in terms of top-1 accuracy than all other models; hence, we can conclude that HLM outperforms state-of-the-art models in handling data inconsistency.

## 5. Discussion

This paper proposes a new mechanism to generate the embedding for predicting next code tokens. This new mechanism is different from all existing frameworks for language models, and can be used in multi-modal or multi-head mechanism, to integrate the existing old mechanisms together, to improve the model performance. Thus, the newly proposed mechanism is meaningful and can be used as a supplement to existing models. To understand the significance of this work, we need to explain the multi-modal mechanism, or the ensemble learning mechanism, to explain why this newly proposed framework has important significance from a research perspective. In the field of machine learning, ensemble learning has been widely used in various scenarios. For non-deep learning models, they often share different data formats, so they can only be integrated in the final step about voting for the final results. For deep learning models, the data formats in different deep learning models are nearly all the same, that data format is the tensor. Then, the ensemble methods in deep learning systems are more advanced and are called multi-modal, multi-view, or multi-head mechanisms. The multi-modal or multi-head mechanisms have been widely used to improve the model efficiency in deep learning systems. Because the data formats in all deep learning models are tensors, we can easily feed the tensors generated by different deep learning models into a deep neural network, to generate a new tensor, this new tensor can be taken as the embedding generated by combining different deep learning models. The more different the integrated multiple deep learning models are, the more information the resulting tensor contains, which ultimately improves the ability of the model to fit data. Once a different encoding or decoding mechanism is proposed, this new mechanism can be combined with the existing old mechanism, to jointly improve the model efficiency. As the proposed mechanism in this paper is different from all existing mechanisms, thus, it is worthy to spend time to combine the mechanism proposed in this paper with other existing mechanisms, to predict the next code in the future.

The interesting finding for this work is that, through the proposed framework, the tree structure of the source code can be used to reduce the data inconsistency between training and testing. Besides, by capturing the tree structure of code, during training, the number of rounds required to reach the convergence state will also be greatly reduced. The existing works predict tokens one by one. The predicting procedure is also named as the decoding procedure. Even for transformer models, the tokens are also predicted (decoded) one by one. When decoding, the information of already visited tokens will be accumulated in the decoding order. In this paper, we find that the information of already visited tokens can be accumulated in a different order than the traditional decoding order. In fact, based on the tree hierarchy, we use an order opposite to the decoding order to collect information about the nodes that have been visited. The procedure of collecting information of already visited tokens is named as the encoding procedure, and the encoding procedure has been separated from the decoding procedure. This is the key scientific contribution, that the encoding process and the decoding process are separated in a language model. This feature is different from existing language models, including transformers. As far as we know, existing models use the same procedure to encode the sequence and decode the sequence. The framework also shows that the encoding procedure can use tree models, and the decoding procedure can use the traditional sequential model. Note that, the encoding procedure can also use other language models, such as the transformer or the graph model. The decoding procedure can also use other language models, such as the transformer, the graph model, or the tree model. Selecting different models for the encoding procedure and the decoding procedure can have different prediction effects. It is worth performing a lot of experiments to find the best model configuration for the encoding procedure and the decoding procedure, although this will consume a lot of code work. In the future, the pre-training method based on transfer learning could also be combined with the framework proposed in this article, to improve the prediction effect.

## 6. Conclusions

The Hierarchical Language Model (HLM) is proposed, to handle the hierarchical structure of the code syntax tree. According to our experiments, the use of the HLM results in an improvement of at least 7% in top-1 accuracy, compared with the LSTM model. The proposed HLM method models the tree structure of the code. To be precise, the method takes the brotherhood of nodes, and the parent–child connections between nodes, into consideration. However, there are still other node relationships that need to be considered. For example, the control–flow or data–flow relationships between statements should be considered. A great deal of engineering implementation is required to accurately extract these complex relationships. In future work, we will adopt more advanced program analysis technology to extract the various relationships of the code, and model these relationships in a deep learning system, to further improve the prediction performance.

**Funding:** This research was funded by National Natural Science Foundation of China (NSFC) grant number 62102220 and the APC was funded by [NSFC-Youth 62102220].

**Institutional Review Board Statement:** Not applicable.

**Informed Consent Statement:** Not applicable.

**Data Availability Statement:** The source code is available at https://github.com/GrowingCode/FrameTokenMemAtten (accessed on 16 February 2023). The data set is available at https://github.com/GrowingCode/CodeCorpus (accessed on 16 February 2023).

**Conflicts of Interest:** The author declares no conflict of interest.

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
