# Peer review of "Improving Code Completion by Solving Data Inconsistencies in the Source Code with a Hierarchical Language Model"

_electronics, doi:10.3390/electronics12071576_

Round 1

Reviewer 1 Report

Title: Improving Code Completion by Solving Data Inconsistency between Training and Testing with Hierarchical Language Model

The authors have developed an improved code completion by solving data inconsistency between training and testing with a hierarchical language model. In a way, the practical analysis used supports the presented framework due to my own observation, and the paper is also relevant to this journal. However, the author has to look into the following concerns:

1.      The title should be looked into and the author should try to avoid the use of abbreviations in the title for example IoT.

2.      The motivation and contribution of this paper should be stated more clearly in the abstract to better understand from the beginning of the study. Authors are advised to be precise in the abstract, and structure your abstract as follows- 1) Background 2) Aim/Objective 3) Methodology 4) Results 5) Conclusion. Write 2-4 lines for each and merge everything in one paragraph (200-300 Words) without any subheading.

1.      The Introduction section should be improved by adding more recent works in this area and providing a more accurate and informative literature review with the pros and cons of the available approaches and how the proposed method is different comparatively.

2.      The related work section is very small, an updated and complete literature review should be conducted and should appear in section 2- Related Work. Some latest papers which studied similar effects problems can be discussed to help the readers.

3.      The figures presented are not too clear, authors should please work on their figures for better quality. For examples figure 1 and 4 is not readable.

4.      The authors need to further emphasize their contributions and relate with the results obtained as is why the adopted methods used are better than the others.

5.      Furthermore, the authors fail in explaining the details of their approach in a clear manner. The overall description and details regarding how the authors derive the result are not clear. The Proposed Method details are given without sufficient explanation and only a few variables appearing in them are addressed, leaving a large part of those obscure.

6.      The author seems to disregard or neglect some important findings in the results that have been achieved in the paper. So elaborate and explain the results in more detail.

7.      It would be interesting for the author explains the limitations of the present study to help other authors for future studies. Mention the future scope of your current works.

8.      Although the English are generally quite good, there are quite a few minor grammatical errors, and a careful read-through is needed to eliminate these errors. The spelling mistake should be corrected by reading through the manuscript.

I appreciate the style of presentation of this paper, but the author needs to incorporate the above-mentioned points for a better and possible publication in the forthcoming conference. I, therefore, recommend a major revision.

Author Response

1. I have slightly changed the title to avoid abbreviation. 
2. The abstract and introduction have been entirely rewritten, I think the modifed version matches the writing framework in the comment. 
3. To explain Figure 1 and Figure 4, I add Figure 2 to show the idea behind the proposed framework. The newly added Figure 2 should be able to answer your doubts. If not, please give me the opportunity for the next round of revision, and I will continue to improve. 
4. I have added two new subsections:  State-of-the-Art Code Completion Method and  Differences and Insights between Existing Models and the Hierarchical Language Model  to explain the base lines in details and show the differences between the proposed framework and the base lines. The newly added two subsections make up two pages. 
5. In addition to the two new paragraphs, other paragraphs describing the new approach have been basically rewritten, which should be much easier to read this time. 
6. I have added a new perspective in experiments: the rounds to converge to show the benefit of the new framework. 
7. I mention the limitation in conclusion. 
8. I have paid $349 to use the MDPI English Editing Service to edit my paper. 

Reviewer 2 Report

The paper addressed a state-of-the-art research area in code analysis. The author presents an improvement to the LSTM method. 

The data collection is all based on JAVA. Although AST can handle different languages, it might be a further study on mix language project?

I would like to know if the author can provide the training time comparison?

Does the author tries to investigate those false prediction and provide some insights about why the false prediction happened?

I also notice some grammar mistakes, for example,

"if the noise unseen data is distributed" should be "if the noise unseen data are distributed".

At the end, I would expect the author to improve the writing on the section 3. Some more examples might help.

Overall, this study shows some interesting results based on author's model and it is worthy for others in similar field to know the author's approach.

Author Response

  1. it is okay to do experiments only on Java because in code completion domains, a lot of related works listed in the Related Work section only do experiments on Java.
  2. I have added the training time statistics and analysis for base lines and the proposed model in experiments. 
  3. The false positive happens mainly due to the limitation of current deep learning systems. The deep learning system can only remember the a fixed length of data, if data is too long, the deep learning system fails to do prediction, this have been proved in many related works. 
  4. I have added two new subsections:  State-of-the-Art Code Completion Method and  Differences and Insights between Existing Models and the Hierarchical Language Model to explain the base line in details and show the differences between the proposed framework and the base lines. The newly added two subsections make up two pages. 
  5. In addition to the two new subsections, other subsections describing the new framework have been basically rewritten, which should be much easier to read this time. 

Reviewer 3 Report

No references form the las 4 years (the latest one is from 2019)

One of the main problems of the paper is that the author presents a discussion with no reference to previous researches.

Check the English of the full document. There are many typos.

Rewrite lines 72 to 81. They are difficult to read.

Some acronyms in the text are not defined the first time they appear: LSTM, UNK, etc.

Introduce in a clearer way the experiment.

Indicate some advantages for the society in general, especially for non-specialists.

What do you mean with the word Vocabulary in table 1.

Indicate units in Table 2. Indicate the meaning of the acronyms.

Line 370 to 386 are not clear, or they seem no to correspond to Table 2.

Indicate units in table 5 and 6.

Author Response

  1. I have added more recent related works and show that our work can be  complementry to the more recent related works. 
  2. I have reorganized the article structure and almost rewritten the abstract and introduction. I hope the paper is more easy to read this time. 
  3. I added two new subsections:  State-of-the-Art Code Completion Method and  Differences and Insights between Existing Models and the Hierarchical Language Model  in to explain the base lines in details and show the differences between the proposed framework and the base lines. The newly added two sections make up two pages. 
  4. In addition to the two new paragraphs, other paragraphs describing the new framework have been basically rewritten, which should be much easier to read this time. 
  5.  For LSTM, UNK, I add the full name in the paper. 
  6. Table 1 just shows the meta information of the projects in data set. Table 1 is just to give readers an overall impression of the dataset, such as how big the dataset is, how many unique words the dataset contains, etc. Table 2 shows the prediction accuracy on projects. Actually, there is no relation between Table 1 and Table 2. I have deleted the content unrelated to the experiments. 
  7. I have paid $349 to use the MDPI English Editing Service to edit my paper. I hope this paper is easier to read this time. Besides, I have tried my best to rewritten this paper, about half of the paper had been entirely written. 

Round 2

Reviewer 1 Report

Although the English are generally quite good, there are quite a few minor grammatical errors, and a careful read-through is needed to eliminate these errors.

The discussion subsection is too small for this kind of paper, authors should try to give details of what the study is all about with details results obtained. Also, the comparative difference between this work and the existing one can be explained. 

Author Response

  1. I have added the nearly half page about the discussion.
  2. In discussion, the significance of the framework and the differences between the proposed framework and the existing frameworks are discussed more carefully. 
  3. Besides, in previous subsection Differences and Insights between Existing Models and the Hierarchical Language Model, the details have been described. 
  4. We only change the subsection Discussion, the other parts are only slightly changed and those parts have been edited by MDPI English Editing Serivce.